# Red Rice Bran Extract Attenuates Adipogenesis and Inflammation on White Adipose Tissues in High-Fat Diet-Induced Obese Mice

**DOI:** 10.3390/foods11131865

**Published:** 2022-06-24

**Authors:** Narongsuk Munkong, Piyanuch Lonan, Wirinya Mueangchang, Narissara Yadyookai, Vaiphot Kanjoo, Bhornprom Yoysungnoen

**Affiliations:** 1Department of Pathology, School of Medicine, University of Phayao, Phayao 56000, Thailand; 2Traditional Chinese Medicine Program, School of Public Health, University of Phayao, Phayao 56000, Thailand; piyanuch.tcm@gmail.com; 3Applied Thai Traditional Medicine Program, School of Public Health, University of Phayao, Phayao 56000, Thailand; wirinya.no@up.ac.th; 4Research and Academic Services, School of Medicine, University of Phayao, Phayao 56000, Thailand; narissara1509@gmail.com; 5Agriculture Program, School of Agriculture and Natural Resources, University of Phayao, Phayao 56000, Thailand; vaiphot.ka@up.ac.th; 6Division of Physiology, Department of Preclinical Science, Faculty of Medicine, Thammasat University, Pathum Thani 12120, Thailand

**Keywords:** red rice bran, obesity, anti-adipogenesis, anti-hypertrophy, anti-inflammation

## Abstract

Red rice bran extract (RRBE) has been reported to have the potential for in vitro metabolic modulation and anti-inflammatory properties. However, little is known about the molecular mechanisms of these potentials in adipose tissue. This study aimed to evaluate the in vivo anti-adipogenic, anti-hypertrophic, and anti-inflammatory activities of RRBE and its major bioactive compounds in mice. After six weeks of consuming either a low-fat diet or a high-fat diet (HFD), 32 mice with initial body weights of 20.76 ± 0.24 g were randomly divided into four groups; the four groups were fed a low-fat diet, a HFD, a HFD plus 0.5 g/kg of RRBE, or a HFD plus 1 g/kg of RRBE, respectively. The 6-week treatment using RRBE reduced HFD-induced adipocyte hypertrophy, lipid accumulation, and inflammation in intra-abdominal epididymal white adipose tissue (*p* < 0.05) without causing significant changes in body and adipose tissue weight, which reductions were accompanied by the down-regulated expression of adipogenic and lipid metabolism genes, including CCAAT/enhancer-binding protein-alpha, sterol regulatory element-binding protein-1c, and hormone-sensitive lipase (*p* < 0.05), as well as inflammatory genes, including macrophage marker F4/80, nuclear factor-kappa B p65, monocyte chemoattractant protein-1, tumor necrosis factor-alpha, and inducible nitric oxide synthase (*p* < 0.05), in adipose tissue. Furthermore, RRBE significantly decreased serum tumor necrosis factor-alpha levels (*p* < 0.05). Bioactive compound analyses revealed the presence of phenolics, flavonoids, anthocyanins, and proanthocyanidins in these extracts. Collectively, this study demonstrates that RRBE effectively attenuates HFD-induced pathological adipose tissue remodeling by suppressing adipogenesis, lipid dysmetabolism, and inflammation. Therefore, RRBE may emerge as one of the alternative food products to be used against obesity-associated adipose tissue dysfunction.

## 1. Introduction

Obesity, especially abdominal or central obesity, has become a major public health concern and is associated with an increased risk of chronic non-communicable diseases, such as dyslipidemia, type 2 diabetes, and cardiovascular diseases [1]. Excessive fat accumulation in intra-abdominal or visceral white adipose tissue (vWAT) is the main pathological characteristic feature of obesity, and it results from both an increase in the size (hypertrophy) and/or number of adipocytes (hyperplasia) [2]. Moreover, enhanced adipogenesis in adipose tissue results in the generation of lipid-laden mature adipocytes from preadipocytes, which requires the induction of numerous transcription factors, such as CCAAT/enhancer-binding protein-alpha (C/EBP-α), peroxisome proliferator-activated receptor-gamma (PPAR-γ), and sterol regulatory element-binding protein-1c (SREBP-1c) [3,4,5]. Adipocytes then become dysfunctional and produce pro-inflammatory mediators through the activation of inflammatory signaling pathways, especially nuclear factor-kappa B (NF-κB), which results in the recruitment of inflammatory cells such as macrophages to adipose tissue, which forms a crown-like structure (CLS) [6,7,8]. Pro-inflammatory mediators, such as tumor necrosis factor-alpha (TNF-α), released by adipocytes and inflammatory cells further contribute to the impairment of insulin signaling in adipocytes and the development of systemic metabolic abnormalities, including the release of free fatty acids (FFAs) from adipocytes through the activation of lipolytic enzymes [9]. Therefore, the mechanisms involved in white adipose tissue (WAT) dysfunction could represent a potential therapeutic target for the treatment of obesity-associated disorders.

Red rice (*Oryza sativa* L.) and its bran are abundant sources of bioactive compounds, such as phenolic compounds, flavonoids, anthocyanins, proanthocyanidins, protocatechuic acid, ferulic acid, γ-oryzanol, and vitamin E [10,11,12,13,14]. Red rice bran extract (RRBE) has been reported to show various beneficial properties, including antioxidant [10], metabolic [11], and anti-inflammatory activities [12,15], which may occur due to the presence and combination of phenolic compounds, such as anthocyanins and proanthocyanidins. A previous study using in vitro chemical-based assays has shown that RRBE from the Hawm or Hawm Dowk Mali Deang variety exhibits the highest phenolic content and antioxidant activities in comparison with the extracts from other colored rice varieties [10]. In vitro studies have also demonstrated that treatment with RRBE enhances glucose uptake and expression of insulin-signaling pathway genes in adipocytes [11] and that treatment with red rice polar extract fraction suppresses the production of pro-inflammatory mediators in macrophages [15], suggesting that RRBE may have beneficial effects on adipose tissue. Additionally, other rice varieties or rice bran extracts, such as black rice and rice bran enzymatic extract, could attenuate adipogenesis in adipocytes [16] and/or hallmarks of adipose tissue dysfunction, such as adipocyte hypertrophy and inflammation, in animals with diet-induced obesity [17,18]. Based on the aforementioned studies, we hypothesized that RRBE might be able to ameliorate obesity-linked WAT dysfunction by attenuating adipogenesis, adipocyte hypertrophy, and inflammation. However, the effects of RRBE on WAT dysfunction have yet to be elucidated. Therefore, the present study aimed to investigate the anti-adipogenic, anti-hypertrophic, and anti-inflammatory activities of RRBE in the WATs of high-fat diet (HFD)-induced obese mice.

## 2. Materials and Methods

### 2.1. Preparation of Ethanol Extracts from Red Rice Bran Samples

Red Hawm or Hawm Dowk Mali Deang rice (red rice) was purchased from Mae Chai Agricultural Cooperative Ltd., Phayao Province, Thailand. The rice bran (1 kg) was mixed in 6 L of 50% ethanol solution (*v*/*v*) at a sample-to-solvent ratio of 1:6 (*w*/*v*) and occasionally stirred for 72 h at room temperature, as previously explained [10,15], albeit with some modifications. Then, the mixture was filtered through filter paper (Whatman No. 1) using vacuum filtration. Following the same methodology, the residue from the mixture was re-extracted twice. The extractant solution was then concentrated under a vacuum using a rotary evaporator. The concentrated extract was then freeze-dried to obtain a dry crude extract. All dried extracts were stored at −20 °C for further analyses.

### 2.2. Analysis of Bioactive Compounds in RRBE

The phenolics and their subtype contents in RRBE were determined as previously described [10,13], with slight modifications. The Folin–Ciocalteu colorimetric method was used to quantify the total phenolic content in the extract, and the results were expressed as mg gallic acid equivalent (GAE) per g extract in dry weight. The total flavonoid and proanthocyanidin contents were determined using aluminum trichloride hexahydrate colorimetric and vanillin acid methods, respectively, and were expressed as mg catechin equivalent (CE)/g extract. The total anthocyanin content was determined by the pH differential method using cyanidin-3-glucoside as a standard and was presented as μg cyaniding-3-glucoside equivalent (C-3-GE)/g extract.

### 2.3. Animal Study

All experiments using mice as subjects were approved by the Institutional Animal Care and Use Committee of the University of Phayao, Thailand (ethical approval code: 59 01 04 0037). Thirty-two male Institute of Cancer Research (ICR) mice (4 weeks old and weighing 20.76 ± 0.24 g; Siam International Co., Ltd., Bangkok, Thailand) were housed in groups of four per cage in an environmentally controlled room (22–25 °C with 60% humidity and a 12 h:12 h light–dark cycle). After a week of acclimatization, the mice were weighed and measured for glucose from tail tip bleeds after overnight fasting using a glucometer, and then fed either a low-fat diet (LFD; 10% kcal from fat, Research Diets Inc., Brunswick, NJ, USA) or a HFD (45% kcal from fat, Research Diets Inc.) for 6 weeks to induce obesity. After this period, the obese mice were then randomized based on body weight and fasting blood glucose into 3 groups that were fed a HFD alone (H), a HFD combined with 0.5 g/kg/day RRBE (HFD + R0.5), or a HFD combined with 1 g/kg/day RRBE (HFD + R1), respectively, for 6 weeks. Meanwhile, the mice fed on the LFD were kept on the same diet for 6 more weeks and assigned to the LFD group. The mice in the RRBE-treated group were administered RRBE dissolved in distilled water once daily by oral gavage, while the mice in the LFD and HFD groups were administered the same volume of distilled water. The animals’ body weight, food consumption, and energy intake were recorded weekly throughout the study. After the end of the treatment period, all the mice were anesthetized with isoflurane in a chamber and their blood samples were collected. Their vWATs were collected from epididymal, retroperitoneal, and mesenteric regions and weighed. The weights were then summed to estimate the total vWAT weight. The relative adipose tissue weight was expressed as adipose tissue weight per final body weight of each mouse (percentage of body weight). Previous studies have found that, in HFD-fed mice, epididymal white adipose tissues (eWATs) show higher adipocyte hypertrophy and inflammation than subcutaneous adipose tissues [19] and that eWATs have a more inflammatory phenotype than retroperitoneal and mesenteric adipose tissues [20]. vWATs have also been shown to be more lipolytically active than subcutaneous adipose tissue [21]. Thus, eWATs were selected and stored at −80 °C or fixed in 10% formalin at room temperature for further analyses.

### 2.4. Histological Analysis

After fixation, the eWATs were embedded in paraffin, sliced at 5 μm thickness, and stained with hematoxylin and eosin (H&E). Histological images were captured using a light microscope (Olympus BX53-P polarizing microscope, Olympus Corporation, Tokyo, Japan) coupled with a digital camera (Olympus DP21 digital microscopy camera, Olympus Corporation). To determine the average adipocyte area, 100 adipocytes/mouse from 6 mice per group were measured in 10 random ×10 microscopic fields, using the AxioVision AC microscopy software version 4.3 (Carl Zeiss, Jena, Germany). The number of CLSs, a hallmark of obesity-associated adipose tissue inflammation defined by the crown- or ring-like accumulation of inflammatory cells such as macrophages surrounding adipocytes, was manually counted in different 10 × 10 fields for 6 mice per group and presented as CLSs/field.

### 2.5. Determination of Lipids in eWATs

The triglyceride (TG) and FFA contents in the eWATs were measured as previously described [5,22], with some modifications. Briefly, approximately 50 mg of eWATs were cut into small pieces and homogenized in 1 mL of isopropanol. The homogenized tissues were mixed and centrifuged at 8000× *g* rpm for 15 min. The lipid levels in the supernatant were determined using commercially available kits (the TG Assay Kit, Erba Diagnostics Mannheim GmbH, Mannheim, Germany, and the FFA Assay Kit, Fujifilm Wako Pure Chemical Corporation, Osaka, Japan) and were normalized to the total levels of proteins in each sample as determined using a Quick Start Bradford Protein Assay Kit (Bio-Rad Laboratories, Inc., Hercules, CA, USA). The results were expressed as µg/mg protein.

### 2.6. Gene Expression Analysis

Total RNA from eWAT was extracted using Trizol reagent (Invitrogen™, Thermo Fisher Scientific Inc., Waltham, MA, USA) as per the manufacturer’s protocol. The total RNA concentration was measured using a NanoDrop 2000 spectrophotometer (Thermo Fisher Scientific Inc.) at 260 nm. Isolated RNA was subsequently reverse-transcribed to cDNA using the cDNA reverse transcription kit (Applied Biosystems™, Thermo Fisher Scientific Inc.), and quantitative real-time polymerase chain reaction (RT-PCR) was performed with the CFX96 Touch RT-PCR system (Bio-Rad Laboratories, Inc.) using TaqMan Gene Expression Master Mix containing gene-specific TaqMan probes (Applied Biosystems™, Thermo Fisher Scientific Inc.). The relative mRNA expression levels were analyzed using the 2^−ΔΔCT^ method by normalizing the expression of glyceraldehyde 3-phosphate dehydrogenase (GAPDH) [23]. Gene-specific TaqMan probes (Applied Biosystems™, Thermo Fisher Scientific Inc.) for mRNA expression analysis were shown as follows: F4/80 (Mm00802529_m1), C/EBP-α (Mm00514283_s1), PPAR-γ (Mm01184322_m1), HSL (Mm00495359_m1), NF-κB p65 (Mm00501346_m1), MCP-1 (Mm00441242_m1), TNF-α (Mm00443258_m1), iNOS (Mm00440502_m1), and GAPDH (Mm99999915_g1).

### 2.7. Serum Analysis

Serum was obtained by centrifugation of whole blood at 3000× *g* rpm for 10 min [24]. The serum levels of monocyte chemoattractant protein-1 (MCP-1) and TNF-α were determined using commercial ELISA kits (Sigma-Aldrich, Inc., St. Louis, MO, USA).

### 2.8. Statistical Analysis

IBM SPSS software version 20 (IBM Corp., Armonk, NY, USA) was used for the statistical analysis of the data. Multiple comparisons among the 4 groups were evaluated using one-way analysis of variance (ANOVA) in conjunction with Tukey’s post hoc test, and a difference of *p* < 0.05 was considered to be statistically significant. For the animal study, the data are presented as the means ± standard error of the mean (SEM). For the study of bioactive compounds, the data are presented as the means ± standard deviation (SD) of triplicate experiments.

## 3. Results

### 3.1. Extraction Yield and Bioactive Components of RRBE

An approximately 8.50% yield of crude-dried extract was obtained. Table 1 shows the total phenolic content, flavonoid content, anthocyanin content, and proanthocyanidin content of RRBE.

### 3.2. Effects of RRBE on Parameters Related to the HFD-Induced Obesity Mouse Model

Overall, during the 12-week study period, the food and energy intake of the HFD group were significantly different from those of the LFD group (*p* < 0.05), while there were no significant differences for the HFD + R0.5 and HFD + R1 groups (Figure 1A,B). After 3 weeks of feeding, the untreated HFD-fed mice displayed significantly higher body weights than those fed the LFD throughout the study period (*p* < 0.05), but no significant increase in body weight at 9 and 12 weeks was observed in mice treated with RRBE (Figure 1C,D). As shown in Figure 1E, the relative adipose tissues in both vWAT and eWAT pads were significantly heavier in the HFD-fed mice than in the controls at the end of the experiment (*p* < 0.05). Although there was no difference in the relative adipose tissue weight after a 6-week treatment, this value was slightly lower in RRBE-treated mice than in untreated HFD-fed mice.

### 3.3. Effect of RRBE on the Histological Appearance of the Adipose Tissue in Obese Mice

The adipose tissue obtained from the LFD group exhibited a normal structure of white adipocytes with a smooth thin layer of cytoplasm surrounding single spherical- or oval-shaped lipid droplets, without any inflammatory cell infiltration (Figure 2A). The LFD group also displayed normal adipocyte size distributions, with higher proportions of both small- and medium-sized cells (Figure 2B). In contrast, the adipose tissue obtained from the HFD group showed a significant increase in adipocyte area and a large number of CLSs (*p* < 0.05; Figure 2A–D). Consistent with these observations, a significant increase in macrophage marker gene (F4/80) expression levels were found in eWATs of the HFD-fed mice compared with the LFD-fed mice (*p* < 0.05; Figure 2E). Treatment of mice with 0.5 and 1 g/kg/day of RRBE significantly lowered adipocyte size, the number of epididymal CLSs, and F4/80 expression compared to the HFD group (*p* < 0.05). However, compared with the LFD group, the adipocyte size distribution of RRBE-treated animals shifted toward larger sizes, as shown in Figure 2B.

### 3.4. Effects of RRBE on Adipose Tissue Lipid Content and Adipogenesis- and Lipid Metabolism-Related Gene Expression in Obese Mice

Biochemical analyses showed that, compared to control mice, in eWATs of mice that were fed HFD-only, the TG content (Figure 3A) was significantly increased (*p* < 0.05), whereas the FFA content (Figure 3B) was only slightly increased. In addition, the mRNA expression levels of C/EBP-α, PPAR-γ, SREBP-1c, and hormone-sensitive lipase (HSL) were significantly increased in eWATs of HFD-fed obese mice (*p* < 0.05; Figure 3C–F). Both TG and FFA levels tended to decrease in the RRBE-treated mice, although there were no statistically significant differences compared to untreated HFD-induced obese mice. Moreover, the administration of RRBE in obese mice also markedly decreased C/EBP-α, SREBP-1c, and HSL mRNA levels (*p* < 0.05), but did not significantly affect PPAR-γ mRNA expressions.

### 3.5. Effects of RRBE on Adipose Tissue Inflammation-Related Gene Expression and Serum Inflammatory Mediators in Obese Mice

As shown in Figure 4A–D, the mRNA expression levels of NF-κB p65, MCP-1, TNF-α, and inducible nitric oxide synthase (iNOS) in eWATs were significantly increased in the HFD-only group compared to those in the LFD group (*p* < 0.05). These over-expressions were significantly reduced in the RRBE-treated group when compared to the HFD group (*p* < 0.05). The HFD-only group was also found to have significantly increased serum concentration of TNF-α (*p* < 0.05), which was restored to normal by RRBE treatment (*p* < 0.05; Figure 4F). However, no significant differences in serum MCP-1 levels were observed between the four groups (Figure 4E).

## 4. Discussion

The present study was conducted to investigate the anti-adipogenic and anti-inflammatory activities of RRBE in an obese mouse model. We hypothesized that RRBE might be able to attenuate HFD-induced WAT dysfunction by suppressing adipogenesis and inflammation. Adipogenesis is known to play a role in the regulation of adipocyte morphology and functions, and it occurs through the activation of its transcription factor cascade [3]. Initially, C/EBP-α and PPAR-γ, key adipogenic transcription factors, bind to their specific DNA sequences and promote the expression of downstream adipocyte-specific genes involved in adipocyte phenotype and lipid synthesis, such as the lipogenic transcription factor SREBP-1c, which ultimately leads to the formation of mature adipocytes along with the accumulation of intracellular lipids. In other words, these transcription factors are not only essential for adipogenesis but also for lipid overaccumulation in adipose tissues or adipocyte hypertrophy [25,26]. Several studies in obese animal models have shown that adipogenic and lipogenic processes are stimulated in adipose tissues in response to positive energy balances or overnutrition, which can promote the accumulation of lipids, mainly TGs, in mature adipocytes and thus contribute to hypertrophic adipose tissue expansion [4,5]. In accordance with these studies, we found that HFD intake increased body weight, fat accumulation, and adipocyte size in the intra-abdominal compartment, as well as up-regulated the expression of C/EBP-α, PPAR-γ, and SREBP-1c genes in eWATs (Figure 1C–E, Figure 2A–C, and Figure 3A–E), indicating that HFD successfully induces obesity and its related mechanisms, including adipogenesis and lipogenesis, in WATs in mice. This obesity phenotype is likely due to the higher energy intake observed in all HFD-fed animals throughout our study (Figure 1B). Importantly, the RRBE treatment attenuated the accumulation of lipids (white adipocyte hypertrophy), and down-regulated the expression of C/EBP-α and SREBP1-c genes in the adipose tissues induced by the HFD, without affecting PPAR-γ gene expression. Treatment with RRBE also slightly tended to reduce adipose tissue and body weight more in obese mice. Our results thus suggest that the anti-adipogenic and anti-lipogenic effects of RRBE may be, at least partly, mediated by the down-regulation of C/EBP-α and SREBP-1c but not PPAR-γ genes, which may contribute to its anti-hypertrophic effects in WATs. However, although RRBE treatment slightly decreased adipose tissue and body weight, as well as reduced the lipid content of the adipose tissues, the differences were statistically insignificant compared to untreated HFD-fed animals, and the percentages of medium and large adipocytes were also higher in RRBE-fed animals than in the controls. In addition, RRBE had no effects on food and energy intake. From these results, we conclude that its beneficial effects are not associated with the regulation of body weight and dietary intake. Our phytochemical screening of RRBE showed the presence of phenolic compounds, including flavonoids, anthocyanins, and proanthocyanidins (Table 1), which have been shown to have beneficial effects, including anti-adipogenic and anti-hypertrophic effects, on adipocytes or adipose tissues [16,27,28,29]. Previous studies have demonstrated that Riceberry rice extract, which contains phenolic compounds, flavonoids, anthocyanins, and other bioactive compounds [16], and anthocyanins from other plant sources, such as fruits of *Vitis coignetiae* Pulliat and black peanut skin [27,28], can inhibit adipogenesis and lipid accumulation in adipocytes by regulating the expression of genes involved in these processes, such as C/EBP-α, PPAR-γ, and SREBP-1c. Meanwhile, proanthocyanidins could reduce visceral adipocyte hypertrophy and improve visceral adipose functionality in cafeteria diet-fed rats, without affecting adiposity and body weight [29]. Moreover, rice bran phenolic extract could inhibit SREBP-1c translocation and lipogenic enzyme expression, thereby decreasing lipid synthesis in the livers of HFD-fed mice [30]. Thus, we infer that the presence of phenolic compounds and flavonoids, including anthocyanins and proanthocyanidins, in RRBE contributes to its anti-adipogenic, anti-lipogenic, and anti-hypertrophic activities. Red rice bran has also been reported to contain other phytochemicals, such as ferulic acid, protocatechuic acid, and γ-oryzanol [10,31], that have suppressive effects on adipogenesis and lipid synthesis in adipocytes [32,33,34], suggesting that these phytochemicals may exert anti-adipogenic, anti-lipogenic, and anti-hypertrophic effects on adipose tissue. Further studies are therefore needed in order to identify the phytochemical constituents responsible for these observed effects of RRBE.

Under conditions of overnutrition and obesity, dysfunctional hypertrophic adipocytes secrete MCP-1 and other chemokines that recruit immune cells, particularly macrophages, for adipose tissue and form CLSs around dead adipocytes [6,7]. These adipocytes and recruited immune cells continuously synthesize chemokines and pro-inflammatory mediators, such as TNF-α, interleukin-6 (IL-6), and nitric oxide, which further promote local and systemic inflammatory responses through the activation of various signaling pathways, including the NF-κB pathway [6,8]. The NF-κB pathway is activated by a variety of stimuli, including cytokines and FFA. It results in the translocation of NF-κB (e.g., p65/p50 dimer) into the nucleus and subsequent transcription of its target genes, such as MCP-1, TNF-α, IL-6, and iNOS [6,7,8,35]. In agreement with previous studies [36,37], HFD feeding resulted in adipose tissue inflammation, as evidenced by increased CLS formation and expressions of macrophage marker gene F4/80, along with increased inflammatory gene expressions, including NF-κB p65, MCP-1, TNF-α, and iNOS, in the HFD group compared to the LFD group (Figure 2D,E and Figure 4A–D). The ELISA results confirmed an increase in the serum levels of TNF-α in mice fed an HFD (Figure 4F), which is one of the main biomarkers of systemic inflammation; this finding is consistent with a previous study [38]. Nevertheless, our result shows that the 12-week HFD feeding duration is not sufficient to increase serum MCP-1 levels (Figure 4E), indicating that this duration is not long enough to induce increased production of this chemokine. This hypothesis is supported by previous findings showing that a significant elevation in serum MCP-1 levels can be observed in mice after a longer period of HFD feeding [39]. However, the aforementioned results indicate that HFD consumption successfully induced both local adipose tissue and systemic inflammation in our diet-induced obesity mouse model. This model is, therefore, suitable for studying the anti-inflammatory activities of RRBE. Interestingly, RRBE significantly reduced HFD-induced inflammatory cell infiltration in eWATs, which was accompanied by decreased expression of macrophage markers, NF-κB p65, and MCP-1. Thus, RRBE can inhibit the infiltration of inflammatory cells, especially macrophages, into the WATs of obese mice by suppressing the expressions of NF-κB and MCP-1. Consistent with these findings, the expression of other inflammatory genes, TNF-α and iNOS, in eWATs, and serum TNF-α concentrations were also attenuated by treatment with RRBE. The current results suggest that the anti-inflammatory effect of RRBE on WAT might be mediated through the down-regulated expression of NF-κB target genes. These results are in agreement with previous findings showing that red rice polar extract fraction, which is rich in phenolic compounds and proanthocyanidins, inhibited TNF, IL-6, and nitric oxide production in lipopolysaccharide-induced Raw 264.7 macrophages by suppressing the NF-κB and other inflammatory signaling pathways [15]. This anti-inflammatory effect is likely due to phenolic compounds, especially proanthocyanidins. Likewise, rice bran phenolic extract supplementation could ameliorate alcohol-induced liver inflammation in mice by reducing the expression of macrophage-related makers, as well as blocking the NF-κB inflammatory pathway [40]. Additionally, other phenolic compounds, including cyanidin-3-O-glucoside (anthocyanin), protocatechuic acid, and ferulic acid, suppress the NF-κB-mediated inflammatory response and the overproduction of pro-inflammatory mediators in lipopolysaccharide-conditioned media-treated adipocytes [41]. Another study also documented that rice bran enzymatic extract supplementation resulted in attenuation of macrophage accumulation in the WATs of HFD-fed mice, accompanied by down-regulation of TNF-α and other inflammatory genes [18]. In concordance with the other studies, the present study suggests that the anti-inflammatory effects of RRBE can probably be attributed to its phenolic compounds, especially proanthocyanidins and anthocyanins (Table 1).

Dysregulation of FFA metabolism in adipose tissue represents an important mechanism linking obesity to cardiometabolic complications [42,43]. Pro-inflammatory cytokines released by dysfunctional obese adipose tissues can impair insulin signaling, which promotes lipolysis through the activation of adipose triglyceride lipase and HSL and inhibits glucose uptake in adipocytes [9,43,44]. This impaired anti-lipolytic action of insulin contributes to increased levels of circulating FFA and subsequent ectopic lipid accumulation in non-adipose tissues, which, in turn, promote both inflammatory and insulin-resistant states, eventually leading to cardiometabolic complications. We found that the expression of HSL, a rate-limiting lipolytic enzyme, was down-regulated in the adipose tissues of RRBE-treated obese mice (Figure 3F), together with a trend toward the reduction of adipose tissue FFA levels (Figure 3B). These results were further supported by our preliminary experiment showing that RRBE reduced FFA levels in the serum of the obese mice (Appendix A). All these data clearly demonstrate that RRBE has an FFA-lowering effect, and its mechanism may be involved in the regulation of HSL gene expression in WATs of HFD-fed mice, which ultimately may improve the impaired regulation of lipolysis. In accordance with our data, a previous study showed that treatment of adipocytes with RRBE, which contains phenolic compounds, flavonoids, anthocyanins, and proanthocyanidins, effectively increased expression levels of the genes encoding insulin-signaling components, such as insulin receptor substrate, Akt2, and glucose transporter [11]. Furthermore, anthocyanin-enriched rice extract [16] and other bioactive phytochemicals, such as protocatechuic acid [41], have been found to inhibit the expression or activation of HSL in adipocytes. The results from this study shed light on the possible effect of RRBE against lipid dysregulation and insulin resistance, which should be further investigated.

## 5. Conclusions

The present study involving an obese mouse model highlights, for the first time, the beneficial effects of RRBE against HFD-induced pathological changes in WATs via the down-regulation of genes involved in adipogenesis, lipogenesis, lipolysis, and inflammation, indicating that RRBE has the potential to serve as a food or dietary supplement to mitigate adipose tissue dysfunction and its associated complications. However, there are some limitations to this study. The direct effects of each active phytochemical compound from RRBE on adipose tissue were not investigated. Therefore, further studies are needed to explore the activities of these active phytochemical compounds.

## Figures and Tables

**Figure 1 foods-11-01865-f001:**
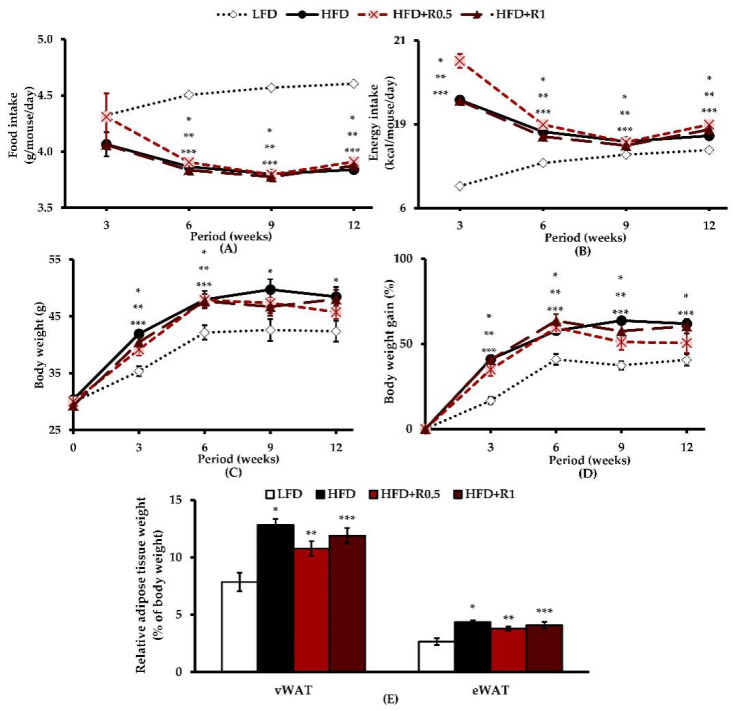
Effects of RRBE on characteristics of HFD-induced obese mice: (**A**) food intake–time curve; (**B**) energy intake–time curve; (**C**) body weight–time curve; (**D**) body weight gain–time curve; (**E**) relative adipose tissue weight. Data are presented as means ± SEM of 8 mice per group and were analyzed by one-way ANOVA followed by Tukey’s post hoc test. *, **, *** *p* < 0.05 for HFD, HFD + R0.5, and HFD + R1 groups versus LFD group, respectively.

**Figure 2 foods-11-01865-f002:**
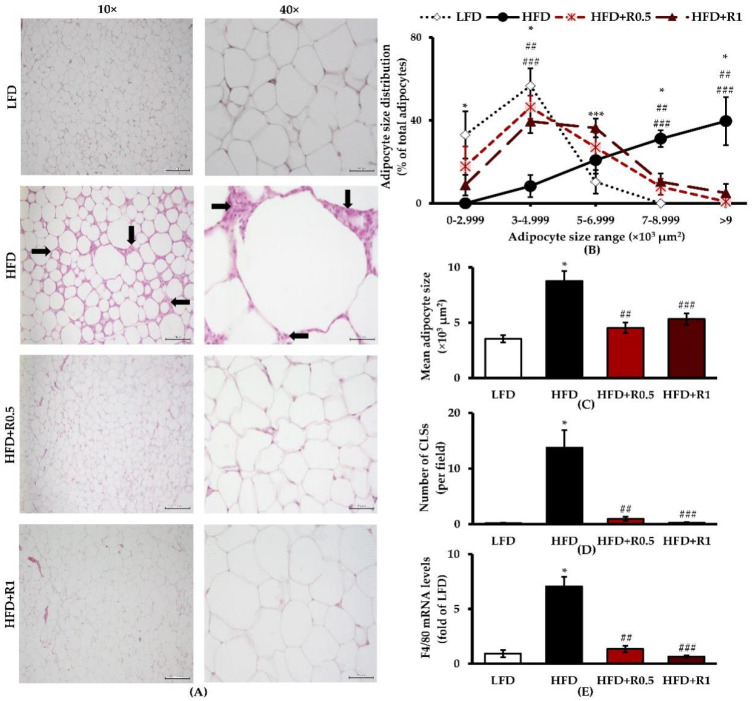
Effects of RRBE on adipose tissue morphology and inflammatory cell accumulation in HFD-fed mice: (**A**) representative microscopic pictures of H&E staining in eWATs; arrows indicate representative pictures of CLSs; left column, magnification 10×; scale bar = 200 μm; right column, magnification 40×; scale bar = 50 μm; (**B**) adipocyte size distribution; (**C**) mean adipocyte size; (**D**) number of CLSs in eWATs; (**E**) F4/80 mRNA expression. Data are presented as means ± SEM of 6 mice per group and were analyzed by one-way ANOVA followed by Tukey’s post hoc test. *, *** *p* < 0.05 for HFD and HFD + R1 groups versus LFD group; ##, ### *p* < 0.05 for HFD + R0.5 and HFD + R1 groups versus HFD group, respectively.

**Figure 3 foods-11-01865-f003:**
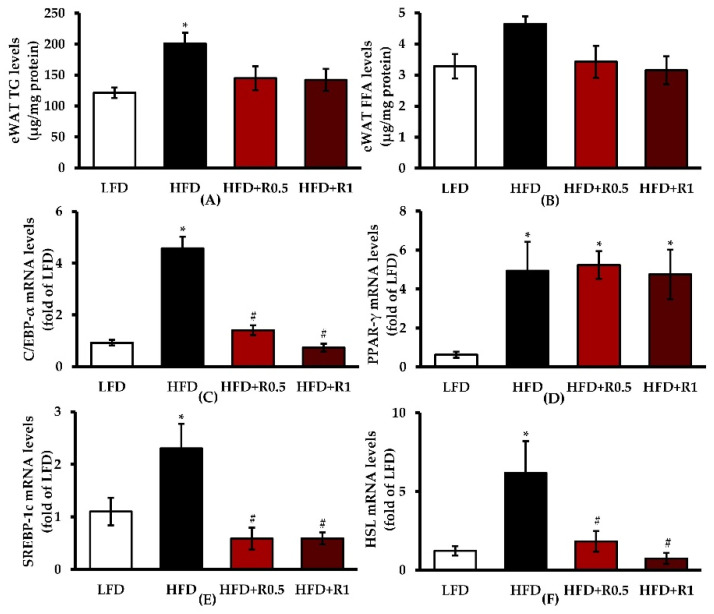
Effects of RRBE on lipid accumulation and adipogenic and lipid metabolism gene expression in eWATs of HFD-fed mice: (**A**) TG content in eWATs; (**B**) FFA content in eWATs; (**C**) C/EBP-α mRNA expression; (**D**) PPAR-γ mRNA expression; (**E**) SREBP-1c mRNA expression; (**F**) HSL mRNA expression. Data are presented as means ± SEM of 6–8 mice per group and were analyzed by one-way ANOVA followed by Tukey’s post hoc test. * *p* < 0.05 versus LFD group; # *p* < 0.05 versus HFD group.

**Figure 4 foods-11-01865-f004:**
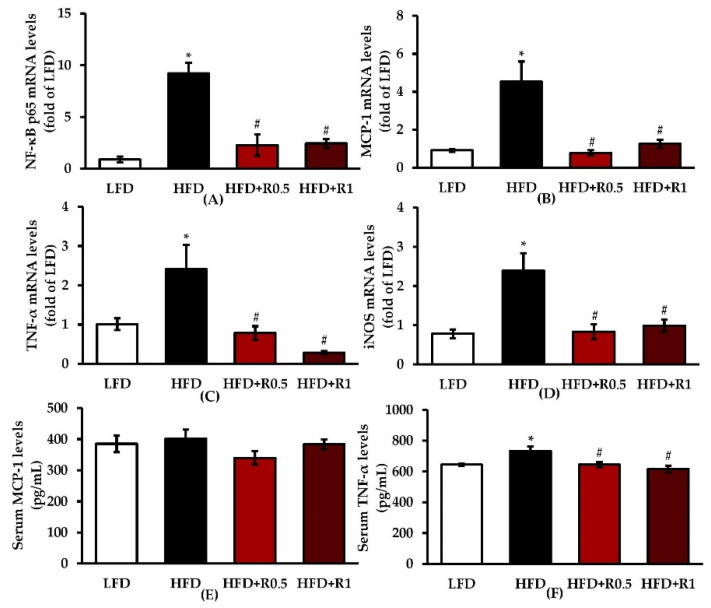
Effects of RRBE on pro-inflammatory gene expression in eWATs and serum inflammatory mediators of HFD-fed mice: (**A**) NF-κB p65 mRNA expression; (**B**) MCP-1 mRNA expression; (**C**) TNF-α mRNA expression; (**D**) iNOS mRNA expression; (**E**) MCP-1 levels in the serum; (**F**) TNF-α levels in the serum. Data are presented as means ± SEM of 6–8 mice per group and were analyzed by one-way ANOVA followed by Tukey’s post hoc test. * *p* < 0.05 versus LFD group; # *p* < 0.05 versus HFD group.

**Table 1 foods-11-01865-t001:** Bioactive compounds in RRBE.

Bioactive Compounds	Content
Phenolics (mg GAE/g)	326.85 ± 7.52
Flavonoids (mg CE/g)	82.84 ± 5.44
Anthocyanins (μg C-3-GE/g)	24.20 ± 13.70
Proanthocyanidins (mg CE/g)	71.49 ± 5.92

Data are presented as means ± SD of triplicate experiments.

## Data Availability

The data presented in this study are available in article.

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
