# Peer review of "Red Rice Bran Extract Attenuates Adipogenesis and Inflammation on White Adipose Tissues in High-Fat Diet-Induced Obese Mice"

_foods, 2022, doi:10.3390/foods11131865_

Round 1

Reviewer 1 Report

Overall comments

The structure of the manuscript is clear, with an appropriate English level. Therefore, it is easy to follow the reading and understand the conveyed message. 

The authors present a proposal for using red rice bran extract as an anti-adipogenic, anti-hypertrophic, and anti-inflammatory agent. The novelty lies in the red rice, which is high in anthocyanins.

As the most important conclusion, I highlight that the RRBE treatment attenuated the accumulation of lipids and white adipocyte hypertrophy and down-regulated the expression of C/EBP- and SREBP1-c genes in the adipose tissues induced by the HFD without affecting PPAR- gene expression.

Specific comments

Introduction 

-In line 78, the author said that: little is known about RRBE's effects on WAT dysfunction. What about other rice bran extract (non-red) effects on WAT dysfunction?. Has it been studied in other kinds of rice? Or is this the first time this type of study has been done with this type of extract? That would increase, if so, the novelty of the work.

-Although an objective is mentioned in the abstract, the objective or hypothesis of the work is not explicitly clear in the introduction. Therefore, at the time of the discussion and conclusion, it is not clear if the research objective was answered. 

Materials and Methods

-In line 139, the authors mention for the first time CLSs, but did not add the meaning. I supposed it referred to crown-like structures. So add the meaning of the acronym.  

-In Gene Expression Analysis, authors should add information about the selected gene targets and primers characteristics (e.g., sequences). 

Do you calculate primer efficiencies? It is necessary to use the 2−ΔΔCT method because it assumes 100% efficiency, and this is not always the case. 

Results

-Line 235: The authors mention normal structure. However, What is the parameter of normal? Could it be a structure characteristic of a healthy adipocyte?

Figure 2b is difficult to understand. It has many data and is tiny. I recommend using a histogram type chart, changing the colors, or reducing the range.

Discussion

Line 339, Authors mentioned: activation of the transcription factor cascade.

Is it one cascade specific o you might want to say: the activation of a transcription factor cascade? 

Lines 369-371: Our phytochemical screening of RRBE showed 369 the presence of phenolic compounds, including flavonoids, anthocyanins, and proanthocyanidins (Table 1), which have been shown to have beneficial effects on adipocytes or adipose tissues 

What are the beneficial effects?

Lines 450-452: Why not present the data in supplementary material? This sentence does not add much because we cannot see the results. Otherwise, it would be better not to mention it. 

Author Response

Introduction

Point 1: In line 78, the author said that: little is known about RRBE's effects on WAT dysfunction. What about other rice bran extract (non-red) effects on WAT dysfunction?. Has it been studied in other kinds of rice? Or is this the first time this type of study has been done with this type of extract? That would increase, if so, the novelty of the work.

Although an objective is mentioned in the abstract, the objective or hypothesis of the work is not explicitly clear in the introduction. Therefore, at the time of the discussion and conclusion, it is not clear if the research objective was answered

Response 1: We thank you for all useful comment and suggestion. We have re-written this point in the introduction section L81-90 as follows:

Additionally, other rice varieties or rice bran extracts, such as black rice and rice bran enzymatic extract, could attenuate adipogenesis in adipocytes [16] and/or hallmarks of adipose tissue dysfunction, such as adipocyte hypertrophy and inflammation, in animals with diet-induced obesity [17,18]. Based on the aforementioned studies, we hypothesized that RRBE might be able to ameliorate obesity-linked WAT dysfunction by attenuating adipogenesis, adipocyte hypertrophy, and inflammation. However, the effects of RRBE on WAT dysfunction are yet to be elucidated. Therefore, the present study aimed to investigate the anti-adipogenic, anti-hypertrophic, and anti-inflammatory activities of RRBE in the WATs of high-fat diet (HFD)-induced obese mice.

Materials and Methods

Point 2: In line 139, the authors mention for the first time CLSs, but did not add the meaning. I supposed it referred to crown-like structures. So add the meaning of the acronym.   

Response 2: We added the information about the meaning of CLSs in the sentence in the material and method section L150-154 as follows:

The number of CLSs, a hallmark of obesity-associated adipose tissue inflammation defined by the crown- or ring-like accumulation of inflammatory cells such as macrophages surrounding adipocytes, was manually counted in different 10 ×10 fields from 6 mice per group and were presented as CLSs/field.  

Point 3: In Gene Expression Analysis, authors should add information about the selected gene targets and primers characteristics (e.g., sequences).

Do you calculate primer efficiencies? It is necessary to use the 2−ΔΔCT method because it assumes 100% efficiency, and this is not always the case. 

Response 3:  We added the TaqMan ID number for each TaqMan probes in the material and method section L176-181 as follows:

Gene-specific TaqMan probes (Applied Biosystems™, Thermo Fisher Scientific Inc.) for mRNA expression analysis were shown as follows: F4/80 (Mm00802529_m1), C/EBP-α (Mm00514283_s1), PPAR-γ (Mm01184322_m1), HSL (Mm00495359_m1), NF-κB p65 (Mm00501346_m1), MCP-1 (Mm00441242_m1), TNF-α (Mm00443258_m1), iNOS (Mm00440502_m1), and GAPDH (Mm99999915_g1). In this study, we used the Taqman gene expression assay which designed the gene-specific probes and primers and TaqMan Gene Expression Master Mix (Applied Biosystems™, Thermo Fisher Scientific Inc.). This Taqman gene expression assay with ready-to-use primer and probe sets  provides only TaqMan ID number, but does not provide the sequences of the primer. The Taqman assay has been optimized and proven by the company for its assay performance and also amplicon efficiency which is on average 100% (+/- 10%).  Moreover, TaqMan probes were selected according to previous published evidences [Montrose et al., 2011; Halade et al., 2011], and gene expression analysis was performed according to the manufacturer’s instructions as well as our previously published assays [Munkong  et al., 2016]. Detailed information for TaqMan probes, specific methods, and manufacturer's manual can be found at https://www.thermofisher.com/th/en/home/brands/applied-biosystems.html.

Montrose, D. C.; Horelik, N. A.;  Madigan, J. P., Stoner, G. D.; Wang, L. S.; Bruno, R. S.; Park, H. J.; Giardina, C.; Rosenberg, D. W. Anti-inflammatory effects of freeze-dried black raspberry powder in ulcerative colitis. Carcinogenesis. 2011, 32, 343–350.

Halade, G.V.; El Jamali, A.; Williams, P.J.; Fajardo, R.J.; Fernandes, G. Obesity-mediated inflammatory microenvironment stimulates osteoclastogenesis and bone loss in mice. Experimental gerontology, 2011, 46, 43–52.

Munkong, N.; Hansakul, P.; Yoysungnoen, B.; Wongnoppavich, A.; Sireeratawong, S.; Kaendee, N.; Lerdvuthisopon, N. Vasoprotective effects of rice bran water extract on rats fed with high-fat diet. Asian Pac J Trop Biomed. 2016, 6, 778-784.

Results

Point 4: Line 235: The authors mention normal structure. However, What is the parameter of normal? Could it be a structure characteristic of a healthy adipocyte?

Response 4: We have re-written this point in the result section L287-291 as follows:

The adipose tissue obtained from the LFD group exhibited a normal structure of white adipocytes with a smooth thin layer of cytoplasm surrounding a single spherical- or oval-shaped lipid droplets, without any inflammatory cell infiltration (Figure 2A). LFD group also displayed normal adipocytes size distribution with the higher proportions of both small- and medium-sized cells (Figure 2B).

Point 5: Figure 2b is difficult to understand. It has many data and is tiny. I recommend using a histogram type chart, changing the colors, or reducing the range.

Response 5: We adjusted the Figure 2B to address the reviewer’s comment.  

Discussion

Point 6: Line 339, Authors mentioned: activation of the transcription factor cascade.

Is it one cascade specific o you might want to say: the activation of a transcription factor cascade?

Response 6: Yes, this transcription factor cascade is specific to the adipogenetic process. We made changes in the disussion section L368-370 as follows: 

Adipogenesis is known to play a role in the regulation of adipocyte morphology and functions, and it occurs through the activation of its transcription factor cascade.

Point 7: Lines 367-370: Our phytochemical screening of RRBE showed 369 the presence of phenolic compounds, including flavonoids, anthocyanins, and proanthocyanidins (Table 1), which have been shown to have beneficial effects on adipocytes or adipose tissues.

What are the beneficial effects?

Response 7: We added the information about beneficial effects in this sentence in discussion section L402-403 to address the reviewer’s concern as follows:

Our phytochemical screening of RRBE showed the presence of phenolic compounds, including flavonoids, anthocyanins, and proanthocyanidins (Table 1), which have been shown to have beneficial effects, including anti-adipogenic and anti-hypertrophic effects, on adipocytes or adipose tissues [16,27-29].

Point 8:.Lines 450-452: Why not present the data in supplementary material? This sentence does not add much because we cannot see the results. Otherwise, it would be better not to mention it.

Response 8: We added the supplementary material (figure S1) to address the reviewer’s comment:

These results were further supported by our preliminary experiment showing that RRBE reduced the FFAs in the serum of the obese mice (Figure S1).

Supplementary Materials: The following supporting information can be downloaded at:www.mdpi.com/xxx/s1, Figure S1: Effects of RRBE on serum FFA levels in HFD-fed mice.

Figure S1. Effects of RRBE on serum FFA levels in HFD-fed mice. Data are presented as mean ± SEM of 4 mice per group and analyzed by one-way ANOVA followed by Tukey’s post hoc test. *p < 0.05 versus LFD group; #p < 0.05 versus HFD group.

Reviewer 2 Report

The authors have investigated an extract from red rice bran and found that it reduces high-fat diet-induced white adipose tissue inflammation and a reduction/normalization of adipocyte size.

The authors suggest that the active ingredients responsible for this biological effect are various potentially anti-oxidant compounds, like phenolics, flavonoids and anthocyanins. One compound that has been shown to reduce adipocyte inflammation, influence WAT-BAT differentiation and other markers of type 2 diabetes is abscisic acid (ABA), which is present in dormant plant seeds. Many of the effects the authors describe from their extract resemble those biological functions. Some sources claim that brown rice have high levels of ABA. Is this compound present in your bran extract?

Author Response

Point 1: The authors have investigated an extract from red rice bran and found that it reduces high-fat diet-induced white adipose tissue inflammation and a reduction/normalization of adipocyte size.

The authors suggest that the active ingredients responsible for this biological effect are various potentially anti-oxidant compounds, like phenolics, flavonoids and anthocyanins. One compound that has been shown to reduce adipocyte inflammation, influence WAT-BAT differentiation and other markers of type 2 diabetes is abscisic acid (ABA), which is present in dormant plant seeds. Many of the effects the authors describe from their extract resemble those biological functions. Some sources claim that brown rice have high levels of ABA. Is this compound present in your bran extract?

Response 1: We thank the reviewer for this suggestion. Red rice bran phytochemicals such as phenolic compounds, anthocyanins, and proanthocyanidins have been reported to have suppressive effects on adipogenesis and lipid synthesis in adipocytes and anti-inflammatory effects on macrophages, suggesting that these phytochemicals may exert their beneficial effects on obesity-associated metabolic and inflammatory diseases. Therefore, our study focused on the investigation of these phytochemicals in RRBE. This study showed that ethanol extract of red rice bran had high content of phenolic compounds, flavonoids, anthocyanins, and proanthocyanidins. It also had non-phenolic phytochemicals such as γ-oryzanol (data not shown). Unfortunately, other bioactive phytochemicals have not been identified in this study. Thus, conducting more studies in the future is necessary to determine other bioactive compounds including abscisic acid in RRBE.

Reviewer 3 Report

Regarding the MS entitled '' Red Rice Bran Extract Attenuates Adipogenesis and Inflammation on White Adipose Tissues in High-Fat Diet-Induced Obese Mice'' I have some comments:

L26. Please define the initial weight of mice and the total number of animals

General comment to abstract. Please add p-value in each significant point.

L40. Please dampen this sentence.

Introduction

L67. Please cite https://doi.org/10.3390/ani11123410

L81. Add hypothesis

L85. Add ref.

L107. Add ± SE or SEM

L107. The company should be in parenthesis

L152. Please add a table show the primers sequence of the studied genes

Figure 1. adipose tissue weight, should be expressed as a percent relative to live body weight.

Figure 2. adipocyte size distribution, please present in in more clear view.

L336. There is a typing error with space, please correct

L371. Mention the beneficial effects

L387. Tissue

Author Response

Point 1: L26. Please define the initial weight of mice and the total number of animals.

General comment to abstract. Please add p-value in each significant point.

Response 1: We would like to thank you for your valuable suggestions for improvement. We added this information in the abstract L22-42 to address the reviewer’s concern as follows:

After six weeks of consuming either a low-fat diet or a high-fat diet (HFD), 32 mice with initial body weights of 20.76 ± 0.24 g were randomly divided into four groups; the four groups were fed a low-fat diet, an HFD, an HFD plus 0.5 g/kg of RRBE, or an HFD plus 1 g/kg of RRBE, respectively. The 6-week treatment using RRBE reduced HFD-induced adipocyte hypertrophy, lipid accumulation, and inflammation in intra-abdominal epididymal white adipose tissue (p < 0.05) without causing significant changes in the body and adipose tissue weight, which were accompanied by down-regulated expressions of adipogenic and lipid metabolism genes including CCAAT/enhancer-binding protein-alpha, sterol regulatory element-binding protein-1c, and hormone-sensitive lipase (p < 0.05), as well as of inflammatory genes including macrophage marker F4/80, nuclear factor-kappa B p65, monocyte chemoattractant protein-1, tumor necrosis factor-alpha, and inducible nitric oxide synthase (p < 0.05), in adipose tissue. Furthermore, RRBE significantly decreased serum tumor necrosis factor-alpha levels (p < 0.05). Bioactive compound analyses revealed the presence of phenolics, flavonoids, anthocyanins, and proanthocyanidins in this extract. Collectively, this study demonstrates that RRBE effectively attenuates HFD-induced pathological adipose tissue remodeling by suppressing adipogenesis, lipid dysmetabolism, and inflammation. Therefore, RRBE may emerge as one of the alternative food products against obesity-associated adipose tissue dysfunction.

Point 2: L40. Please dampen this sentence.

Response 2: We have re-written this point in the abstract L41-42 as follows:

Therefore, RRBE may emerge as one of the alternative food products against obesity-associated adipose tissue dysfunction.

Point 3: L67. Please cite https://doi.org/10.3390/ani11123410.

Response 3: We added this reference (14) in reference section.

Point 4: L81 Add hypothesis.

Response 4: We added our hypothesis in the introduction section L84-87 as follows:

Based on the aforementioned studies, we hypothesized that RRBE might be able to ameliorate obesity-linked WAT dysfunction by attenuating adipogenesis, adipocyte hypertrophy, and inflammation. However, the effects of RRBE on WAT dysfunction are yet to be elucidated. Therefore, the present study aimed to investigate the anti-adipogenic, anti-hypertrophic, and anti-inflammatory activities of RRBE in the WATs of high-fat diet (HFD)-induced obese mice.

Point 5: L85. Add ref.

Response 5: We added the references (10,15) in material and method section L96-97 as follows:

The rice bran (1 kg) was mixed in 6 L of 50% ethanol solution (v/v) at a sample-to-solvent ratio of 1:6 (w/v) and occasionally stirred for 72 hours at room temperature, as previously explained [10,15], albeit with some modifications.

Point 6: L107. Add ± SE or SEM and the company should be in parenthesis

Response 6: We added the SEM and the company in parenthesis in material and method section L 117 as follows:

Thirty-two male Institute of Cancer Research (ICR) mice (4 weeks old and weighing 20.76 ± 0.24 g; Siam International Co., Ltd., Bangkok, Thailand) were housed in groups of four per cage in an environmentally controlled room (22–25 °C with 60% humidity and 12 h:12 h light–dark cycle).

Point 7: L152. Please add a table show the primers sequence of the studied genes

Response 7: We added the TaqMan ID number for each TaqMan probes in the Material and Method section L176-181 as follows:

Gene-specific TaqMan probes (Applied Biosystems™, Thermo Fisher Scientific Inc.) for mRNA expression analysis were shown as follows: F4/80 (Mm00802529_m1), C/EBP-α (Mm00514283_s1), PPAR-γ (Mm01184322_m1), HSL (Mm00495359_m1), NF-κB p65 (Mm00501346_m1), MCP-1 (Mm00441242_m1), TNF-α (Mm00443258_m1), iNOS (Mm00440502_m1), and GAPDH (Mm99999915_g1).

In this study, we used the Taqman gene expression assay which designed the gene-specific probes and primers and TaqMan Gene Expression Master Mix (Applied Biosystems™, Thermo Fisher Scientific Inc.). This Taqman gene expression assay with ready-to-use primer and probe sets  provides only TaqMan ID number, but does not provide the sequences of the primer. The Taqman assay has been optimized and proven by the company for its assay performance and also amplicon efficiency which is on average 100% (+/- 10%).  Moreover, TaqMan probes were selected according to previous published evidences [Montrose et al., 2011; Halade et al., 2011], and gene expression analysis was performed according to the manufacturer’s instructions as well as our previously published assays [Munkong  et al., 2016]. Detailed information for TaqMan probes, specific methods, and manufacturer's manual can be found at https://www.thermofisher.com/th/en/home/brands/applied-biosystems.html.

Montrose, D. C.; Horelik, N. A.;  Madigan, J. P., Stoner, G. D.; Wang, L. S.; Bruno, R. S.; Park, H. J.; Giardina, C.; Rosenberg, D. W. Anti-inflammatory effects of freeze-dried black raspberry powder in ulcerative colitis. Carcinogenesis. 2011, 32, 343–350.

Halade, G.V.; El Jamali, A.; Williams, P.J.; Fajardo, R.J.; Fernandes, G. Obesity-mediated inflammatory microenvironment stimulates osteoclastogenesis and bone loss in mice. Experimental gerontology, 2011, 46, 43–52.

Munkong, N.; Hansakul, P.; Yoysungnoen, B.; Wongnoppavich, A.; Sireeratawong, S.; Kaendee, N.; Lerdvuthisopon, N. Vasoprotective effects of rice bran water extract on rats fed with high-fat diet. Asian Pac J Trop Biomed. 2016, 6, 778-784.

Point 8: Figure 1. adipose tissue weight, should be expressed as a percent relative to live body weight.

Response 8: We adjusted the Figure 1E to address the reviewer’s comment. 

Point 9: Figure 2. adipocyte size distribution, please present in in more clear view.

Response 9: We adjusted the Figure 2B to address the reviewer’s comment. 

Point 10: L336. There is a typing error with space, please correct

Response 10: We corrected the a typing error with space.

Point 11: L371. Mention the beneficial effects

Response 11: We added the information about beneficial effects in this sentence in discussion section L402-403 to address the reviewer’s concern as follows:

Our phytochemical screening of RRBE showed the presence of phenolic compounds, including flavonoids, anthocyanins, and proanthocyanidins (Table 1), which have been shown to have beneficial effects, including anti-adipogenic and anti-hypertrophic effects, on adipocytes or adipose tissues [16,27-29].

Point 12: L387. Tissue

Response 12: We corrected this word in dicussion section L419.

Red rice bran has also been reported to contain other phytochemicals such as ferulic acid, protocatechuic acid, and γ-oryzanol [10,31] that have suppressive effects on adipogenesis and lipid synthesis in adipocytes [32-34], suggesting that these phytochemicals may exert anti-adipogenic, anti-lipogenic, and anti-hypertrophic activities on adipose tissue.       

Round 2

Reviewer 3 Report

Dear authors

Thank you for the revisions. The manuscript was improved